# Impact of extent of internal acoustic meatus tumor removal using translabyrinthine approach for acoustic neuroma surgery

Kuan-Wei Chiang[1,2], Sanford P. C. Hsu[2,3], Tsui-Fen Yang[2,4], Mao-Che Wang[1,2,5]*

1 Department of Otorhinolaryngology-Head and Neck Surgery, Taipei Veterans General Hospital, Taipei, Taiwan, 2 School of Medicine, National Yang Ming Chiao Tung University, Taipei, Taiwan, 3 Departments of Neurosurgery, Neurological Institute, Taipei Veterans General Hospital, Taipei, Taiwan, 4 Department of Physical Medicine and Rehabilitation, Taipei Veterans General Hospital, Taipei, Taiwan, 5 Institute of Hospital and Health Care Administration, National Yang Ming Chiao Tung University, Taipei, Taiwan

* wangmc@vghtpe.gov.tw

**Data Availability Statement:** All relevant data are contained as Supporting Information files.

**Funding:** The authors received no specific funding for this work.

## Abstract

### Objectives

Many studies have investigated the surgical outcome and predictive factors of acoustic neuroma using different approaches. The present study focused on large tumors due to the greater likelihood of internal acoustic meatus involvement and the greater application of surgical intervention than radiosurgery. There have been no previous reports on outcomes of internal acoustic meatus tumor removal. We investigated the impact of the extent of internal acoustic meatus tumor removal using a translabyrinthine approach for large acoustic neuroma surgery and predictive factors of tumor control.

### Methods

This retrospective study reviewed 104 patients with large cerebellopontine angle tumor >3 cm treated by translabyrinthine approach microsurgery. Predictive factors of postoperative facial palsy, tumor control, and extent of internal acoustic meatus tumor removal were assessed.

### Results

The mean tumor size was 38.95 ± 6.83 mm. Postoperative facial function showed 76.9% acceptable function (House–Brackmann grade 1 or 2) six months after surgery. The extent of internal acoustic meatus tumor removal was a statistically significant predictor factor of poor postoperative facial function. Younger age, larger tumor size needing radiosurgery, and more extensive removal of tumor were associated with better tumor control.

### Conclusion

More extensive internal acoustic meatus tumor removal was associated with poor postoperative facial function and better tumor control.

**Competing interests:** The authors have declared that no competing interests exist.

## Introduction

Most tumors in the cerebellopontine angle are benign and slow-growing [1–3]. Vestibular schwannoma, also known as acoustic neuroma, is the most commonly reported type of cerebellopontine angle tumor [1–3]. Due to the characteristics of these tumors, the routine treatment protocols are observation, stereotactic radiotherapy, and surgical intervention including the middle cranial fossa approach, suboccipital/retrosigmoid approach, and translabyrinthine approach [1, 3, 4]. Suitable management for each individual case involves careful consideration of factors including patient age, health status, tumor size and location, residual hearing in both ears, and neurological symptoms due to tumor compression [3, 5]. In 1894, Sir Charles Ballance was the first to successfully resect an acoustic neuroma. Since then, patient morbidity and mortality rates have reduced due to a gradual refining of the approach technique [3, 6]. Given the functional and cosmetic effects of facial nerve dysfunction, it is important to determine the perioperative factors that influence the preservation of facial nerve function. Larger tumor size with greater possibility of internal acoustic meatus involvement may affect the surgical outcome. There are few reports discussing the extent of internal acoustic meatus tumor removal. Surgery aims to completely remove the tumor and minimize complications in patients with acoustic neuroma [7, 8]. The modern concept of facial nerve preservation with subtotal and near-total resection is generally established and well understood, and residual tumor has been deemed to be acceptable. However, the tumor remnant should be monitored by regular imaging and controlled with radiosurgery if there is a significant progression in tumor size. Therefore, the present study investigated predictive factors for tumor control.

The present study reviewed the prevalence of postoperative facial palsy after removal of large vestibular schwannoma using translabyrinthine approach microsurgery, predictive factors of postoperative facial palsy, tumor control, and the extent of internal acoustic meatus tumor removal.

## Materials and methods

### Ethical approval

Ethical considerations were made and all data were fully anonymized prior to commencing the study. Consent was waived for this study that is a retrospective chart review. The Institutional Review Board (2) of Taipei Veterans General Hospital granted approval for the study (TPEVGH IRB number: 2019-04-007BC).

### Study design

A retrospective review of patients' medical records was conducted at the department of otorhinolaryngology–head–neck surgery and neurosurgery in a tertiary referral center. All patients who were surgically treated for acoustic neuroma using the translabyrinthine approach microsurgery between June 2011 and November 2019 were included in our study. All operations were performed by a senior neurosurgeon (P.C.H.) in conjunction with a senior neurotologist (M.C.W.).

Patient charts, operative notes, follow-up clinic notes, and radiographic studies were thoroughly reviewed. We focused on lesions >3 cm that were treated using the translabyrinthine approach microsurgery. Patient data were collected from clinical records and imaging data, and included age, sex, tumor size and components, admission duration, grade of preoperative and postoperative facial function, preoperative pure tone audiometry, preoperative and postoperative Gamma Knife radiosurgery (GKRS), tumor with internal auditory canal involvement, percentage of tumor removal, extent of internal acoustic meatus tumor removal,

trigeminal nerve involvement, lower cranial nerve involvement, distance between internal auditory canal and jugular bulb, and postoperative cerebrospinal fluid (CSF) leakage.

We graded preoperative hearing levels on a scale of A to D, according to the classification scheme published by the American Academy of Otolaryngology–Head and Neck Surgery (AAO-HNS) guidelines [9]. Class A was defined as a hearing level of $\leq$ 30 dB and a word recognition score (WRS) of > 70%; class B was defined as a hearing level of 30–50 dB and a WRS of > 50%; class C was defined as a hearing level of>50 dB and a WRS >50%; class D was defined as any hearing level with WRS of < 50%.

All patients underwent presurgical quantifying magnetic resonance imaging (MRI) and high-resolution temporal bone computed tomography (CT). Tumor size was assessed by an otorhinolaryngologist (K.W.C.) by direct measurement using the maximal diameter of the extracanalicular component from the axial/coronal view of MRI scan. The minimal distance between the internal auditory canal and jugular bulb was obtained from the coronal view of a temporal bone CT scan. The percentage of tumor removal was determined by serial MRI and intraoperative findings, then divided into partial, subtotal, near total, and total removal using previously reported scales [4, 10]. We defined the extent of tumor removal as "total removal" when no visible tumor remnant was observed at the end of surgery and on performing serial MRI, "near total removal" when there was a $\leq$ 5% residual tumor, "subtotal removal" when the tumor was removed by 80%–90% of its volume, and "partial removal" when the tumor remnant was more than "subtotal removal". Following MRI, we classified the tumor internal auditory canal involvement into three categories: none, half involvement, and full involvement. Additionally, the extent of internal acoustic meatus tumor removal was defined as none, partial, and total removal.

Facial nerve function was assessed in patients prior to and after operation at six-month follow-up using the House–Brackmann facial nerve outcome scale [11]. All patients aged <20 years, neurofibromatosis type II (NF2), non-vestibular schwannoma, recurrent tumor, preoperative facial palsy grade III–VI, and mortalities were excluded from the study.

GKRS was arranged if regrowth of tumor remnant was identified from subsequent postoperative MRI by the senior neurosurgeon.

## Statistical analysis

Statistical analyses were performed using SPSS version 18 (SPSS Inc. Released 2009. PASW Statistics for Windows, Version 18.0. Chicago: SPSS Inc.). The chi-square and Fisher's exact tests were used for categorical data, and a difference at a probability level <0.05 was considered as significant. Independent $t$ test was applied on continuous data and significance was defined as p < 0.05. Multivariable linear regression was used to assess the relationship between certain predictors of facial paralysis and scales of facial function.

## Results

### Patient characteristics

Among the 111 cases treated using translabyrinthine approach microsurgery enrolled in our study, seven patients with tumor size less <3 cm were excluded. Cases with pathology of neurofibromatosis type II, mortalities, aged <20 years, recurrence, and preoperative facial palsy grade III–VI were excluded at the beginning of the study. A final total of 104 cases were included. All patients had complete medical records and imaging follow-ups at our outpatient clinic.

Patients' demographic data are listed in Table 1. The mean age of the patients was 46.90 ± 13.11 years, with a median age of 48 (range, 22–77) years. Overall, 53% patients were female and 47% patients were male. The average duration of admission was 10.10 ± 4.96 days.

**Table 1. Demographic data of patients.**

| Total cases number | 104 | |
|---|---|---|
| | **Case number** | **%** |
| Sex | | |
| Male | 49 | 47% |
| Female | 55 | 53% |
| Age (Mean±SD) | 46.90 ± 13.11 | |
| Admission duration (day) | 10.10 ± 4.96 | |
| Tumor Component | | |
| Solid | 21 | 20% |
| Heterogeneous | 54 | 52% |
| Cyst | 29 | 28% |
| Tumor size (mm, mean±SD) | 38.95 ± 6.83 | |
| IAC involvement | | |
| 0 | 3 | 3% |
| 0-half canal | 19 | 18% |
| full | 82 | 79% |
| Tumor removal | | |
| partial | 1 | 1% |
| subtotal | 6 | 6% |
| near total | 44 | 42% |
| total | 53 | 51% |
| Extent of IAC tumor removal | | |
| 0 | 4 | 4% |
| Partial | 13 | 13% |
| Total removal | 87 | 84% |
| Facial numbness | | |
| Yes | 67 | 64% |
| No | 37 | 36% |
| Lower cranial nerve involve | | |
| Yes | 18 | 17% |
| No | 86 | 83% |
| Pre-op hearing (dBHL, mean±SD) | 71.21±31.94 | |
| Pre-op radiosurgery | | |
| Yes | 5 | 5% |
| No | 99 | 95% |
| Post-op radiosurgery | | |
| Yes | 10 (*1 case IMRT) | 10% |
| No | 94 | 90% |
| Post-op CSF leakage | | |
| Yes | 12 | 12% |
| No | 92 | 88% |
| IAC-JB distance (mm, mean±SD) | 6.60 ± 2.58 | |

Abbreviations: IAC, internal auditory canal; JB, Jugular bulb; CSF, cerebrospinal fluid; IMRT, Intensity modulated radiation therapy.

More than 87% patients had non-serviceable hearing in the diseased ear. The preoperative hearing level was assessed by pure tone audiometry (average threshold for 0.5, 1, 2, and 4 kHz) and showed a mean preoperative hearing level of 71.21 ± 31.94 dBHL, with 68.3% of patients

categorized as class C and 1.9% of patients categorized as class D according to the AAO-HNS hearing classification guidelines.

Tumor components were classified as solid, heterogeneous, and cyst using preoperative MRI scans [5]. Cystic tumors comprised 28% of the cases, whereas 20% were solid tumors, and 52% were heterogeneous tumors.

The mean tumor size was 38.95 ± 6.83 mm, and the maximum tumor size was 62 mm. Three patients had no internal auditory canal involvement, 19 cases were half-canal involvement (18%), and 82 cases with full involvement (79%) were identified.

Considering tumors with trigeminal nerve involvement, 67 patients (64%) presented with facial numbness. Among these patients, 18 patients (17%) presented with lower cranial nerve involvement, such as uvula or tongue deviation, choking, or vocal palsy, during preoperative physical examinations.

## Facial function preservation

House–Brackmann grade 1 or 2 was defined as good facial nerve function [4, 6, 12, 13]. In all surgeries using the translabyrinthine approach, facial nerve function was anatomically preserved without surgical interruption in all patients, except for tumor involvement at the mastoid segment in two cases (1.9%). In the present study, 80 patients (76.9%) were evaluated with good facial nerve function (House–Brackmann grade 1 or 2) after surgery. Univariate analysis revealed that patient age, gender, tumor size and component, admission duration, tumor involvement of internal auditory canal, preoperative pure tone audiometry, preoperative and postoperative GKRS, percentage of tumor removal, trigeminal nerve invasion, lower cranial nerve involvement, and distance between the internal auditory canal and jugular bulb were not significantly related to facial functional outcomes. However, the extent of internal acoustic meatus tumor removal was a statistically significant predictor factor of postoperative facial function ($p < 0.05$) (Table 2).

## Tumor control

The mean duration of follow up of 104 cases was 39 months, range of duration was 6 to 107 months and median duration was 34.5 months. Periodic MRI scans were performed three days and three months after surgery to monitor tumor regrowth of the residual tumor. We checked MRI every 6 to 12 months afterwards to monitor tumor growth. Ninety-seven percent of patients had total and near-total resection on the surgery. Ten patients (9.6%) required GKRS or radiotherapy for further tumor control (Table 3), indicating a tumor control rate of 90.3%. The mean duration from the initial surgery to GKRS for ten cases was 12 months and the range of duration was 4 to 25 months. There was an increased demand for postoperative radiosurgery or radiotherapy in the younger age group and those with a larger tumor size. Greater extent of gross tumor and internal acoustic meatus tumor removal was significantly related to better tumor control ($p < 0.05$).

No significant differences were observed for other factors, including gender, admission duration, tumor size component, tumor involvement of the internal auditory canal, preoperative pure tone audiometry, preoperative radiotherapy, trigeminal nerve invasion, lower cranial nerve involvement, distance between internal auditory canal and jugular bulb, postoperative facial palsy, and postoperative CSF leakage.

## Discussion

### Predictive factor of postoperative facial palsy

In previous studies, the goals of surgery for cerebellopontine angle tumors have been to relieve tumor compression-related neurological signs [6]. The most common and concerning

**Table 2. Predictive factors of facial palsy (multivariable linear regression).**

|  | All patients (n = 104) | Number (%) of facial palsy | *p* value |
|---|---|---|---|
| Age (mean ± SD) | 46.90 ± 13.11 |  | 0.759 |
| Gender (male: female) | 49:55 |  | 0.384 |
| Admission duration (day) | 10.10 ± 4.96 |  | 0.776 |
| Tumor size (mm) | 38.95 ± 6.83 |  | 0.147 |
| Tumor component |  |  | 0.198 |
| Solid | 21 (20%) | 3 (14%) |  |
| Heterogeneous | 54 (52%) | 11 (20%) |  |
| Cystic | 29 (28%) | 10 (35%) |  |
| IAC involvement |  |  | 0.784 |
| 0 | 3 (3%) | 0 (0%) |  |
| 0.5 | 19 (18%) | 3 (16%) |  |
| Full | 82 (79%) | 21 (26%) |  |
| Tumor removal |  |  | 0.887 |
| Partial | 1 (1%) | 0 (0%) |  |
| Subtotal | 6 (6%) | 2 (33%) |  |
| Near total | 44 (42%) | 8 (18%) |  |
| Total | 53 (51%) | 14 (26%) |  |
| Extent of IAC tumor removal |  |  | **0.037**[**] |
| 0 | 4 (4%) | 0 (0%) |  |
| Partial | 13 (13%) | 1 (8%) |  |
| Total removal | 87 (84%) | 23 (26%) |  |
| Facial numbness |  |  | 0.398 |
| Yes | 67 (64%) | 16 (24%) |  |
| No | 37 (36%) | 8 (22%) |  |
| Lower cranial nerve involve |  |  | 0.741 |
| Yes | 18 (17%) | 5 (28%) |  |
| No | 86 (83%) | 19 (22%) |  |
| Pre-op hearing (dBHL) | 71.21 ± 31.94 |  | 0.266 |
| Pre-op radiosurgery |  |  | 0.591 |
| Yes | 5 (5%) | 2 (40%) |  |
| No | 99 (95%) | 22 (22%) |  |
| Post-op radiosurgery |  |  | 0.127 |
| Yes | 10 (10%) *1 case IMRT | 3 (30%) |  |
| No | 90 (90%) | 21 (22%) |  |
| CSF leakage |  |  | 0.168 |
| Yes | 12 (12%) | 1 (8%) |  |
| No | 92 (88%) | 23 (25%) |  |
| IAC-JB distance (mm) | 6.60 ± 2.58 |  | 0.920 |

Abbreviations: IAC, internal auditory canal; JB, Jugular bulb; CSF, cerebrospinal fluid; IMRT, Intensity modulated radiation therapy.

[**]Values in bold indicate statistically significant (p < 0.05) by linear regression.

complication is postsurgical facial paralysis. Considering that most tumors in the posterior fossa are benign, adequate tumor resection with fewer complications is currently the primary aim of surgery. Cushing proposed that subtotal resection of tumors with refined operative ability could reduce mortality and complications [14]. Tumor size is a known prognostic factor related to preservation of facial nerve function, and smaller tumors tend to have better facial

**Table 3. Predictive factors for further tumor control.**

| | All patients (n = 104) | Post-op GKRS (n = 10) | No post-op GKRS (n = 94) | p value |
|---|---|---|---|---|
| Age (mean ± SD) | 46.90 ± 13.11 | 38.24 ± 12.71 | 47.83 ± 12.88 | **0.027**[**] |
| Gender (male: female) | 49:55 | 3:7 | 46:48 | 0.328 |
| Admission duration (day) | 10.10 ± 4.96 | 12.9 ± 6.15 | 9.8 ± 4.76 | 0.060 |
| Tumor size (mm) | 38.95 ± 6.83 | 44.3 ± 7.23 | 38.38 ± 6.57 | **0.009**[**] |
| Tumor component | | | | 0.125 |
| Solid | 21 (20%) | 0 (0%) | 21 (100%) | |
| Heterogenous | 54 (52%) | 8 (15%) | 46 (85%) | |
| Cystic | 29 (28%) | 2 (7%) | 27 (93%) | |
| IAC involvement | | | | 0.635 |
| 0 | 3 (3%) | 0 (0%) | 3 (100%) | |
| 0.5 | 19 (18%) | 1 (5%) | 18 (95%) | |
| full | 82 (79%) | 9 (11%) | 73 (89%) | |
| Tumor removal | | | | **0.004**[**] |
| Partial | 1 (1%) | 0 (0%) | 1 (100%) | |
| Subtotal | 6 (6%) | 3 (50%) | 3 (50%) | |
| Near total | 44 (42%) | 5 (11%) | 39 (89%) | |
| Total | 53 (51%) | 2 (4%) | 51 (96%) | |
| Extent of IAC tumor removal | | | | |
| 0 | 4 (4%) | 0 (0%) | 4 (100%) | **0.020**[**] |
| Partial | 13 (13%) | 4 (31%) | 9 (69%) | |
| Total removal | 87 (84%) | 6 (7%) | 81 (93%) | |
| Facial numbness | | | | 0.741 |
| Yes | 67 (64%) | 6 (9%) | 61 (91%) | |
| No | 37 (36%) | 4 (11%) | 33 (89%) | |
| Lower cranial nerve involve | | | | 0.372 |
| Yes | 18 (17%) | 3 (17%) | 15 (83%) | |
| No | 86 (83%) | 7 (8%) | 79 (92%) | |
| Facial palsy | | | | 0.693 |
| No (gr 1,2) | 80 (77%) | 7 (9%) | 73 (91%) | |
| Yes (gr 3–6) | 24 (23%) | 3 (12%) | 21 (88%) | |
| Pre-op hearing (dBHL) | 71.21 ± 31.94 | 76.30 ± 31.15 | 70.67 ± 32.14 | 0.599 |
| Pre-op radiosurgery | | | | >0.999 |
| Yes | 5 (5%) | 0 (0%) | 5 (100%) | |
| No | 99 (95%) | 10 (10%) | 89 (90%) | |
| CSF leakage | | | | >0.999 |
| Yes | 12 (12%) | 1 (8%) | 11 (92%) | |
| No | 92 (88%) | 9(10%) | 83 (90%) | |
| IAC-JB distance (mm) | 6.60 ± 2.58 | 4.41 ± 3.52 | 6.90 ± 2.36 | 0.054 |

Abbreviations: GKRS, Gamma Knife radiosurgery; IAC, internal auditory canal; JB, Jugular bulb; CSF, cerebrospinal fluid.

[**]Values in bold indicate statistically significant (p < 0.05) by Chi-square & Fisher's exact tests, independent t test.

function outcomes. Gurgel and Monfared advocated subtotal resection of tumor related to improve preservation of facial nerve [4, 10]. In 2001, Wiet et al. [15] reported that older patients had worse facial paralysis outcomes.

Facial nerve function was preserved from the labyrinthine segment to the mastoid segment during surgery in all patients by a neurotologist. Intraoperative facial nerve monitoring with a

**Table 4.  Review of articles for facial nerve preservation.**

| First author | Tumor size | Number of cases | Surgical approach | Total resection % | Facial n. preservation at last follow-up % |
|---|---|---|---|---|---|
| Lanmann et al. [22] | >3cm | 190 | TL | 96 | 52.6 |
| Briggs et al. [3] | - | 167 | TL | | 42 |
| Wu et al. [2] | >3cm | 40 | TL | 97.5 | 65 |
| Sluyter et al. [23] | >2cm | 120 | TL | 92 | 56 |
| Mamikoglu et al. [16] | >3cm | 81 | TL | 95.1 | 45 |
| Sanna et al. [18] | >3cm | 175 | TL | 85.1 | 29.5 |
| Deveze et al. [11] | Koos IV | 110 | TL | 82 | 60 |
| Brackmann et al. [12] | Mean 2.4cm | 512 | TL | - | 81 |
| Shamji et al. [19] | Mean 2.3cm | 128 | TL | - | 87 |
| Charpiot et al. [17] | >4cm | 123 | TL | | 42 |
| Rinaldi et al. [20] | all size | 97 | TL | 94 | 61.1 |
| Gurgel et al. [10] | >2.5cm | 555 | TL | - | 62.5 |
| Jungmin Ahn et al. [5] | all size | 91 | TL | 70.3 | 70 |

Abbreviations:TL, translabyrinthine approach

physiatrist was applied to all patients to identify the facial nerves and preserved nerve function during tumor dissection.

Postoperative facial function after six months showed 76.9% of patients with normal or near-normal facial nerve function (House–Brackmann Grade 1 or 2), which was strongly predicted by the extent of internal acoustic meatus tumor removal ($p < 0.05$). There was a statistically significant relationship between postoperative facial palsy and extent of internal acoustic meatus tumor removal, indicating an increased risk of facial palsy with more extensive tumor resection of internal acoustic meatus. Our findings suggest that surgeons should take care when manipulating facial nerves while removing tumors in the internal acoustic meatus as well as in the cerebellopontine angle.

A previous review of large acoustic neuroma via the translabyrinthine approach showed that preservation of facial nerve at the time of last follow-up was 29.5 to 65% [2, 3, 5, 10–12, 16–23] (Table 4). Iatrogenic nerve injury could be prevented with the assistance of a facial nerve monitor and advanced surgical techniques [5, 10, 23]. Intraoperative nerve edema and thermal injury during mastoidectomy using the translabyrinthine approach could explain an initial temporary facial paralysis [19, 24]. Anatomical preservation of facial nerves and good teamwork are essential to preserve facial nerve function [12].

## Adequate tumor control

Emphasis on tumor removal with preservation of facial function was the prevailing current trend of surgery for vestibular schwannoma [4]. GKRS was considered a proper salvage treatment modality to achieve better tumor control for postoperative tumor recurrence or progression, with the goal of deferring tumor growth or shrinking the tumor size [25].

In the present study, the total and near-total resection rates were up to 93% in patients that were evaluated by intraoperative findings of the surgeon and periodic MRI. Furthermore, only 9.6% of cases required GKRS or radiotherapy for further tumor control during follow-up. According to our findings, the young patient group had more chances of receiving postoperative GKRS ($p = 0.027$). More aggressive management tended to be introduced for young patients when surgeons assessed tumor regrowth or residue. Tumor size was another factor

that affected tumor control (p = 0.009) since a greater tumor size was associated with a greater likelihood of tumor residue.

In addition to the above factors, greater extent of gross tumor and internal acoustic meatus tumor removal was significantly related to better tumor control (p < 0.05). Although the modern concept of subtotal and near-total resection with remnant for facial nerve preservation is acceptable, it remains unclear how to achieve a balance between tumor control and preserved facial outcome.

On comparison to Bloch's study [26], we had 44 cases with near-total resection and 7 cases with partial and subtotal resection in our study under the same definition of resection extent. In the group of near-total resection, the recurrence rate was 11.4% and lower than 42.9% of subtotal resection group. Our results were poor compared to that of Bloch that reported 3% and 32% recurrence in near-total and subtotal resection, respectively; however, this may be due to large tumor sizes of the cases included in our study. Similar to previous reports, our results support the idea that near-total resection has greater therapeutic outcomes and a lower risk of recurrence. Nonetheless, postoperative follow-ups using serial MRI for the early detection of recurrence are necessary.

## Limitations

The number of cases that we eligible for enrollment to our study was reduced due to our inclusion criteria of large tumor size >3 cm and the focus on acoustic neuroma. In the present study, a six-month follow-up period was sufficient to reveal the results of postoperative facial nerve function, however it may be better to have a longer follow up period to determine definite tumor control.

## Conclusion

The extent of internal acoustic meatus tumor removal using a translabyrinthine approach for large acoustic neuroma surgery was correlated with poor postoperative facial function and better tumor control.

## Supporting information

**S1 File. Dataset.** This is the supporting information of our included cases.
(PDF)

## Acknowledgments

We thank Hsin-Yi Huang from the Biostatistics Task Force, Taipei Veterans General Hospital, for the statistical assistance.

## Author Contributions

**Conceptualization:** Kuan-Wei Chiang, Sanford P. C. Hsu, Mao-Che Wang.

**Formal analysis:** Kuan-Wei Chiang, Mao-Che Wang.

**Investigation:** Kuan-Wei Chiang, Sanford P. C. Hsu, Tsui-Fen Yang, Mao-Che Wang.

**Methodology:** Kuan-Wei Chiang, Sanford P. C. Hsu, Mao-Che Wang.

**Project administration:** Kuan-Wei Chiang, Sanford P. C. Hsu, Tsui-Fen Yang, Mao-Che Wang.

**Supervision:** Kuan-Wei Chiang, Sanford P. C. Hsu, Tsui-Fen Yang, Mao-Che Wang.

**Writing – original draft:** Kuan-Wei Chiang.

**Writing – review & editing:** Sanford P. C. Hsu, Tsui-Fen Yang, Mao-Che Wang.

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
