## [Decision Letter · Decision Letter 0]

13 Jan 2021

PONE-D-20-38944

Impact of extent of internal acoustic meatus tumor removal using translabyrinthine approach for acoustic neuroma surgery

PLOS ONE

Dear Dr. WANG,

Thank you for submitting your manuscript to PLOS ONE. After careful consideration, we feel that it has merit but does not fully meet PLOS ONE’s publication criteria as it currently stands. Therefore, we invite you to submit a revised version of the manuscript that addresses the points raised during the review process.

In addition to the reviewers' comments, please address the following:

Statistically, there is likely a multicollinearity issue.  Extent of IAC removal is unlikely to be independent of extent of IAC involvement.  In addition, extent of IAC removal is unlikely to be independent of overall tumor removal.

For example - might partial IAC involvement be more likely to be completely removed as opposed to a tumor completely filling the canal? 

We look forward to receiving your revised manuscript.

Kind regards,

Jennifer Alyono

Academic Editor

PLOS ONE

2. In the ethics statement in the manuscript and in the online submission form, please provide additional information about the patient records/samples used in your retrospective study, including: a) whether all data were fully anonymized before you accessed them; b) the date range (month and year) during which patients' medical records/samples were accessed; c) the source of the medical records/samples analyzed in this work (e.g. hospital, institution or medical center name).

Reviewers' comments:

Reviewer's Responses to Questions

**Comments to the Author**

1. Is the manuscript technically sound, and do the data support the conclusions?

Reviewer #1: Partly

Reviewer #2: Partly

2. Has the statistical analysis been performed appropriately and rigorously? 

Reviewer #1: Yes

Reviewer #2: I Don't Know

3. Have the authors made all data underlying the findings in their manuscript fully available?

Reviewer #1: Yes

Reviewer #2: Yes

4. Is the manuscript presented in an intelligible fashion and written in standard English?

Reviewer #1: No

Reviewer #2: Yes

5. Review Comments to the Author

Reviewer #1: Manuscript Title:

Impact of extent of internal acoustic meatus tumor removal using translabyrinthine approach for acoustic neuroma surgery

Summary

This study aimed to examine facial nerve function and tumor control in patients undergoing translabyrinthine resection of large CPA/IAC tumors. They found that IAC involvement was significantly associated with worse facial nerve outcomes, and also found several factors such as extent of tumor removal affected rates of tumor control.

Strengths and weaknesses

I think this study was overall thorough in its attempt to find predictive factors for the aforementioned outcomes.

There are many grammatical errors and awkwardly worded sentences in the manuscript that are distracting. Some of these errors (not all) are indicated under Specific Comments below. It may be beneficial for the authors to consult with an English Language proof-reading service.

Additionally, some of their findings, while they show statistical differences are not necessarily novel ideas. The finding that facial nerve outcomes are worse with IAC dissection is not surprising, and does not necessarily need statistical analyses to come to that conclusion.

There were several other issues with the manuscript I have written in the specific comments below.

Specific comments

Line 59: suggest replacing “nonuseful” with “non-serviceable”

Line 92: suggest replacing “facial palsy” with “facial function”

Lines 99-104: Please provide a figure illustrating how the measurements were performed.

Line 104: Please either provide a figure or describe the “previously reported scales”.

Line 104-105: Please rephrase this sentence, it is not clear what the authors are trying to state.

Line 107: would not use the word “invasion” as this suggests a malignant process.

Line 108: what do the authors mean by partial? What percentage of tumor removal would be categorized as “partial”?

Line 112: Were previously radiated patients included in the study?

Line 113-115: Please re-word this sentence.

Lines 117-118: Awkwardly worded sentence, please change.

Lines 125-126: Would remove this sentence.

Lines 137: Please rephrase this sentence.

Line 137-141: Please put a reference to the AAO HNS classification guidelines and either describe what the classes are or place a table showing what this is. Was there a 71 dB average drop in hearing? That would be surprising given that these are all translab approaches. If this it the case, I would recommend reporting the pre-op and post-op discrimination scores.

Table 1:

- Again, what is the difference between subtotal and partial tumor removal The authors do not make this clear.

- “Low” cranial nerve should be “Lower”

Table 3:

- What does GKS stand for?

Lines 155-157: The facial nerve stimulate on the nerve monitor at the end of the case?

Lines 192-193: Please re phrase this sentence.

Lines 203-207: While the authors found a statistically worse facial nerve outcome in cases that involved the IAC, I would not call this a novel idea. This is inferring that more stretching and manipulation of the nerve leads to a worse facial nerve outcome, which isn’t surprising and does not necessarily need statistical tests to know.

Line 225: Would replace the word ‘demands’ in this sentence.

Reviewer #2: The authors report a retrospective review of 104 patients with large vestibular schwannomas undergoing translabyrinthine resection to investigate predictive factors for facial nerve outcomes and tumor control. This is an interesting and relevant topic for skull base surgeons. The stated conclusions are that more extensive internal auditory canal tumor dissection was associated with better tumor control at the expense of worse postoperative facial function.

The study is well done, reported in a logical manner, and presents relevant literature to the discussion. I believe there is merit in this study and the results should be shared, however I have some primary concerns with the methodology and statistical analysis that I would like addressed further. Additionally, the manuscript could benefit from a thorough editing process to catch minor errors, some of which I have highlighted, and to allow for better flow.

I would appreciate if the authors could address the following points, in no particular order of significance:

-What was the mean duration of follow up (and the ranges) used to assess tumor control? Were any tumors observed beyond the three month post-op MRI, or was that the cutoff to assess tumor control? If not, do the authors feel confident that they are capturing all cases of subsequent tumor growth following such a short follow-up period? While the end results may not ultimately change, I would like to see a longer duration of follow-up before reaching the conclusions drawn in the study with regards to tumor control.

-Were any patients treated prophylactically with GKS prior to demonstrating tumor growth?

-It appears that the 3 patients without IAC tumor involvement were included in the full analysis. Was there any consideration to exclude these patients? Why or why not? Should we suppose that IAC tumor involvement is a factor in facial nerve outcome or tumor control rate, or the manipulation (or lack thereof) within the IAC?

-To follow the last point, can the authors please expand the data in Table 2 to reflect the number of patients in each subgroup with good facial nerve function?

-The authors state linear regression was used to determine the significance of various independent variables on facial nerve outcomes and tumor control. Did the authors consider any form of adjusted multivariate regression models to control for various confounding factors? In all tables with p-values, it is suggested that authors denote the type of statistical test used and include in footnotes all adjustments accounted for in the modeling framework, if used.

-Were any patients treated with postoperative steroids? Did this have any impact on facial nerve outcomes?

-The authors report two patients with tumor involvement within the mastoid segment of the facial nerve. This would represent an unusual tumor growth pattern. Could the authors please expand on this further?

-The authors suggest that there are no prior studies regarding the impact of tumor remnant with facial nerve outcomes or recurrence. Did the authors consider the following: Bloch DC, Oghalai JS, Jackler RK, Osofsky M, Pitts LH. The fate of the tumor remnant after less-than-complete acoustic neuroma resection. Otolaryngol Head Neck Surg 2004;130:104-12? How does the present study compare to the findings from prior studies with regards to tumor control based on the degree of tumor removal?

Some stylistic or grammatical considerations (not an exhaustive list):

-The introduction seems non-linear, including some methodological details of the study regarding surgical approach, surgical teams, etc. Consider consolidating these details to the methods section for better flow.

-Line 137: “Most” to “more”? “Healing” to “hearing”

-Line 192-193: Can you please clarify this sentence – these patients were not suggested for what?

Overall, this is a very interesting study that may be strengthened by some additional clarifications and/or revisions to the statistical methods. I encourage the authors to revise and resubmit the manuscript for further review.

6. PLOS authors have the option to publish the peer review history of their article (what does this mean?). If published, this will include your full peer review and any attached files.

Reviewer #1: No

Reviewer #2: **Yes: **Pedrom C. Sioshansi, MD

---

## [Author Response · Author response to Decision Letter 0]

28 Jan 2021

We have revised the manuscript as attachment and suggested to read the document of "response to reviewers" for better text composition and picture of following response. Thank you.

EDITOR SUGGESTIONS:

1. Statistically, there is likely a multicollinearity issue. Extent of IAC removal is unlikely to be independent of extent of IAC involvement. In addition, extent of IAC removal is unlikely to be independent of overall tumor removal. For example - might partial IAC involvement be more likely to be completely removed as opposed to a tumor completely filling the canal? 

• RESPONSE: Thank you very much for your question. Yes, there might be collinearity issue between extent of IAC removal and IAC involvement. And there might also be collinearity issue between extent of IAC removal and overall tumor removal. However, we consider the issue may not exist for the following reasons. First, if there was multicollinearity issue among these three factors. When we put all these three factors in the regression model, we should find no statistics significance in all these three factors. On the other hand, we do see the statistics significance in the factor of extent of IAC removal which showed in table 2. Second, the percentage of facial palsy in partial IAC involvement and total ISC tumor removal was quite different (16% vs 26%) which was shown in revised table 2. Third, we do not see correlation or upward trend between the extent of overall tumor removal and percentage of facial palsy which was shown in revised table 2.

REVIEWER 1 COMMENTS:

1. Line 59: suggest replacing “nonuseful” with “non-serviceable”

RESPONSE: Thank you for this suggestion. We have corrected the words in our revised manuscript. 

2. Line 92: suggest replacing “facial palsy” with “facial function”

RESPONSE: We have corrected the words in our revised manuscript. Thank you for help.

3. Lines 99-104: Please provide a figure illustrating how the measurements were performed.

RESPONSE: We illustrated the measurements of the minimal distance between the internal auditory canal and jugular bulb by one patient’s coronal view of a temporal bone CT scan as following. 

4. Line 104: Please either provide a figure or describe the “previously reported scales”.

RESPONSE: We cited with reference 4 and 8 for extent of tumor removal scales as following: The percentage of tumor removal was determined by serial MR imaging and surgical findings of surgeons. We defined the extent of tumor removal as follows: total removal represented no visible tumor remnant at the end of surgery and serial MR imaging; near total removal was any residual tumor ≤ 5%; subtotal removal when the tumor was resected 80% to 90% by volume; partial removal was tumor remnant more than definition of subtotal removal. We have added this comment to the section of study design. Thank you for this suggestion.

5. Line 104-105: Please rephrase this sentence, it is not clear what the authors are trying to state.

RESPONSE: Thank you for this suggestion. We have removed this sentence because that was redundant after careful consideration.

6. Line 107: would not use the word “invasion” as this suggests a malignant process.

RESPONSE: Thank you for this suggestion. We have revised the word with “involvement” in our revised manuscript. 

7. Line 108: what do the authors mean by partial? What percentage of tumor removal would be categorized as “partial”?

RESPONSE: We defined the extent of IAC tumor removal as none, partial, and total removal. If remnant tumor was found in inner auditory canal by MRI performed 3 days postoperatively, that would be classified to “partial”.

8. Line 112: Were previously radiated patients included in the study?

RESPONSE: Yes, we enrolled 5 cases previously radiated at other hospital but the tumor size was still larger than 3cm. We have recorded as “Pre-op radiosurgery” group at table 1.

9. Line 113-115: Please re-word this sentence.

RESPONSE: Thank you for this suggestion. We have revised the sentence “Stereotactic radiotherapy or radiotherapy” to “gamma-knife radiosurgery (GKRS)” in our revised manuscript. 

10. Lines 117-118: Awkwardly worded sentence, please change.

RESPONSE: Thank you for this suggestion. We have removed this sentence because that was redundant after careful consideration.

11. Lines 125-126: Would remove this sentence.

RESPONSE: Thank you for this suggestion. We have removed this sentence in our revised manuscript. 

12. Lines 137: Please rephrase this sentence.

RESPONSE: Thank you for this suggestion. We have rephrased this sentence as “More than 87% patients had non-serviceable hearing in the diseased ear. Preoperative hearing level was assessed by pure tone audiometry (average threshold for 0.5, 1, 2, and 4 kHz) and showed a mean preoperative hearing level of 71.21 ± 31.94 dBHL” in our revised manuscript. 

13. Line 137-141: Please put a reference to the AAO HNS classification guidelines and either describe what the classes are or place a table showing what this is. Was there a 71 dB average drop in hearing? That would be surprising given that these are all translab approaches. If this it the case, I would recommend reporting the pre-op and post-op discrimination scores.

RESPONSE: We are sorry for misunderstanding and the sentence was rephrased to “preoperative hearing level” instead of dropped hearing. 

14. Table 1:

- Again, what is the difference between subtotal and partial tumor removal The authors do not make this clear.

RESPONSE: We cited with reference 4 and 8 for extent of tumor removal scales as following: The percentage of tumor removal was determined by serial MR imaging and surgical findings of surgeons. We defined the extent of tumor removal as follows: total removal represented no visible tumor remnant at the end of surgery and serial MR imaging; near total removal was any residual tumor ≤ 5%; subtotal removal when the tumor was resected 80% to 90% by volume; partial removal was tumor remnant more than definition of subtotal removal. We have added this comment to the section of study design. Thank you for this suggestion.

15. - “Low” cranial nerve should be “Lower”

RESPONSE: Thank you for this suggestion. We have corrected the word in our revised manuscript. 

16. Table 3:

- What does GKS stand for?

RESPONSE: We are sorry for misunderstanding. The proper word should be “gamma-knife radiosurgery (GKRS)”. We have corrected the word in our revised manuscript. 

17. Lines 155-157: The facial nerve stimulate on the nerve monitor at the end of the case?

RESPONSE: Yes, we checked facial nerve at the end of the case. 

18. Lines 192-193: Please re phrase this sentence.

RESPONSE: Thank you for this suggestion. We have removed this sentence in our revised manuscript. 

19. Lines 203-207: While the authors found a statistically worse facial nerve outcome in cases that involved the IAC, I would not call this a novel idea. This is inferring that more stretching and manipulation of the nerve leads to a worse facial nerve outcome, which isn’t surprising and does not necessarily need statistical tests to know.

RESPONSE: We agreed that postoperative facial palsy might be influenced by manipulation of nerve during operation. In our manuscript, we also suggested that surgeons should take care when manipulating facial nerves while removing tumors in the internal acoustic meatus as well as in the cerebellopontine angle at line 197 to 199 (revised version). We have removed the word “novel” in our revised manuscript. 

20. Line 225: Would replace the word ‘demands’ in this sentence.

RESPONSE: Thank you for this suggestion. We have rephrased the sentence as “According to our findings, the young patient group had more chance to receive post-operative GKRS (p = 0.027)” in our revised manuscript. 

REVIEWER 2 COMMENTS:

1. What was the mean duration of follow up (and the ranges) used to assess tumor control? Were any tumors observed beyond the three month post-op MRI, or was that the cutoff to assess tumor control? If not, do the authors feel confident that they are capturing all cases of subsequent tumor growth following such a short follow-up period? While the end results may not ultimately change, I would like to see a longer duration of follow-up before reaching the conclusions drawn in the study with regards to tumor control.

• RESPONSE: Thank you for providing these insights. The mean duration of follow up of 104 cases was 39 months, the ranges of duration were 6-107 months and the median duration was 34.5 months. We arranged periodic MRI scans within three days and three months after surgery to monitor tumor regrowth of the residual tumor. After then, we checked MRI every 6 to 12 months to assess tumor control. According to our study, the mean duration from operation to receiving GKRS of 10 cases was 12 months, the ranges of duration were 4 to 25 months. Therefore, it can be proved that our duration of follow-up time was enough for surveillance of tumor recurrence. We have supplemented the result with explanations of above data in our revised manuscript.

2. Were any patients treated prophylactically with GKS prior to demonstrating tumor growth?

• RESPONSE: No, all of patients did not treat prophylactically with GKRS. Postoperative GKRS was arranged only when the size of tumor remnant enlarged.

3. It appears that the 3 patients without IAC tumor involvement were included in the full analysis. Was there any consideration to exclude these patients? Why or why not? Should we suppose that IAC tumor involvement is a factor in facial nerve outcome or tumor control rate, or the manipulation (or lack thereof) within the IAC?

• RESPONSE: Thank you for providing these insights. There were 3 patients without IAC tumor involvement who were included in the 4 patients without IAC tumor removal. That means no perioperative manipulation of IAC and facial nerve if patients without IAC tumor involvement. Therefore, extent of IAC tumor removal seems to be more significant factor of our surgical outcome. 

4. To follow the last point, can the authors please expand the data in Table 2 to reflect the number of patients in each subgroup with good facial nerve function?

• RESPONSE: Thank you for this suggestion. We have incorporated your comments by expanding the data in Table 2 in our revised manuscript. 

5. The authors state linear regression was used to determine the significance of various independent variables on facial nerve outcomes and tumor control. Did the authors consider any form of adjusted multivariate regression models to control for various confounding factors? In all tables with p-values, it is suggested that authors denote the type of statistical test used and include in footnotes all adjustments accounted for in the modeling framework, if used.

• RESPONSE: Thank you for this suggestion. We have revised the tables with the type of statistical test used in our revised manuscript. 

6. Were any patients treated with postoperative steroids? Did this have any impact on facial nerve outcomes?

• RESPONSE: You have raised an important point; however, more than 90% patients treated with postoperative steroids for prevention of postoperative brain edema and increased intracranial pressure. Because most of patients treated with postoperative steroids, we consider that would not have impact on our results.

7. The authors report two patients with tumor involvement within the mastoid segment of the facial nerve. This would represent an unusual tumor growth pattern. Could the authors please expand on this further?

• RESPONSE: Thank you for providing these insights. We rechecked the pathology report of these patients with tumor involvement within mastoid segment of the facial nerve that indeed proved as vestibular schwannoma. However, large tumor size of these two cases may have impact of extrusion on nerve. We applied the preoperative MRI from one of the cases for example as follows. 

8. The authors suggest that there are no prior studies regarding the impact of tumor remnant with facial nerve outcomes or recurrence. Did the authors consider the following: Bloch DC, Oghalai JS, Jackler RK, Osofsky M, Pitts LH. The fate of the tumor remnant after less-than-complete acoustic neuroma resection. Otolaryngol Head Neck Surg 2004;130:104-12? How does the present study compare to the findings from prior studies with regards to tumor control based on the degree of tumor removal?

• RESPONSE: We appreciate your comments and drew some comparison with this article. Compared to Bloch’s study, there were 44 cases with near-total resection and 7 cases with partial and subtotal resection in our study under the same definition of resection extent. In the group of near-total resection, the recurrence rate was 11.4% and lower than 42.9% of subtotal resection. Though our results seemed to be worse than Bloch that reported with 3% and 32% recurrence in near-total and subtotal resection separately, large tumor size of our cases may affect this outcome. However, the suppose of near-total resection having greater therapeutic outcome and a lower risk of recurrence was approved not only in previous studies but our results. Postoperative alert surveillance with serial MRI to detect early recurrence is necessary. We have added this comment to the section of discussion. 

9. The introduction seems non-linear, including some methodological details of the study regarding surgical approach, surgical teams, etc. Consider consolidating these details to the methods section for better flow.

• RESPONSE: Thank you for this suggestion. We have modified the methodological details from introduction to the methods section in our revised manuscript.

10. Line 137: “Most” to “more”? “Healing” to “hearing”

• RESPONSE: We agreed with your comments and have corrected the words in our revised manuscript.

11. Line 192-193: Can you please clarify this sentence – these patients were not suggested for what?

RESPONSE: Thank you for this suggestion. We have removed this sentence because that was redundant after careful consideration.

---

## [Decision Letter · Decision Letter 1]

16 Mar 2021

PONE-D-20-38944R1

Impact of extent of internal acoustic meatus tumor removal using translabyrinthine approach for acoustic neuroma surgery

PLOS ONE

Dear Dr. WANG,

Thank you for submitting your manuscript to PLOS ONE. After careful consideration, we feel that it has merit but does not fully meet PLOS ONE’s publication criteria as it currently stands. Therefore, we invite you to submit a revised version of the manuscript that addresses the points raised during the review process.

We look forward to receiving your revised manuscript.

Kind regards,

Jennifer Alyono

Academic Editor

PLOS ONE

Additional Editor Comments (if provided):

Regarding involvement of the mastoid segment: does this mean the mastoid segment of the facial nerve had vestibular schwannoma within it? How is this distinguished from facial nerve schwannoma? The pathology report alone cannot distinguish the nerve of origin.

Reviewers' comments:

Reviewer's Responses to Questions

**Comments to the Author**

1. If the authors have adequately addressed your comments raised in a previous round of review and you feel that this manuscript is now acceptable for publication, you may indicate that here to bypass the “Comments to the Author” section, enter your conflict of interest statement in the “Confidential to Editor” section, and submit your "Accept" recommendation.

Reviewer #1: All comments have been addressed

Reviewer #2: All comments have been addressed

2. Is the manuscript technically sound, and do the data support the conclusions?

Reviewer #1: Yes

Reviewer #2: Partly

3. Has the statistical analysis been performed appropriately and rigorously? 

Reviewer #1: Yes

Reviewer #2: Yes

4. Have the authors made all data underlying the findings in their manuscript fully available?

Reviewer #1: Yes

Reviewer #2: Yes

5. Is the manuscript presented in an intelligible fashion and written in standard English?

Reviewer #1: No

Reviewer #2: Yes

6. Review Comments to the Author

Reviewer #1: While the authors did a nice job correcting some of the issues addressed, there are still a significant amount of grammatical and syntax errors that is highly distracting. I suggest the authors use an English Language proof reading service to ensure that the writing is more streamlined and easier to understand. Please see some of the suggestions below:

Lines 64-65: This sentence needs to be rephrased. There certainly are papers discussing IAC tumor removal. The authors need to make this sentence clearer.

Line 72: ‘Incidence’ should be ‘prevalence’.

Lines 72-75: This sentence should be revised. Facial nerve palsy complicating surgery is awkwardly phrased.

Lines 86: Please consider removing this sentence. It is redundant, you have already stated the translab approach was used in the previous sentence.

Line 89-90: This sentence needs revision, either remove ‘and reports’ or change the sentence to avoid having two ‘and’s’.

Lines 107-109: Please add a figure showing how the measurements were made.

Line 114: ‘surgical findings of surgeons’ is awkward and redundant, should just be ‘intraoperative findings’

Line 125: ‘and other pathological types’ is vague and needs rewording

Lines 127-130: Sentence needs to be rephrased. It is repetitive and awkwardly rephrased.

I’m not sure why there are no line numbers in the paragraphs between lines 155 and 156, but the sentence with the word ‘half-invasion’ needs to be redone.

Line 173: “After then” should be changed

Lines 198-204: This paragraph was already stated in the methods section, unclear why it is in the discussion again

Lines 247-250: This sentence is unclear and needs to be revised.

Lines 249: Please remove the word ‘alert’

Reviewer #2: Thank you to the authors for providing responses to the prior comments and for incorporating the feedback into their manuscript. I believe these changes have strengthened the manuscript significantly, however there remain several concerns I have regarding the conclusions drawn by the authors.

The authors provided additional details regarding the statistical methods used. Given the multiple factors that likely affect surgical outcomes - both facial function and tumor control - I would have liked to see multivariate adjustment in the regressions to determine the significance of individual tumor or treatment factors of interest. If this was already done, it is not clear to the reader and should be clarified.

Additionally, I remain concerned that there is adequate duration of follow-up for all patients to draw the stated conclusion regarding tumor control. Consider that in some reports the average time to remnant tumor growth is over 1 year (Kasbekar AV, Adan GH, Beacall A, Youssef AM, Gilkes CE, Lesser TH. Growth Patterns of Residual Tumor in Preoperatively Growing Vestibular Schwannomas. J Neurol Surg B Skull Base. 2018;79(4):319-324. doi:10.1055/s-0037-1607421), yet some patients in the current study have duration of follow-up as short as 6 months. I commend that the authors have acknowledged this as a limitation, yet they make a (admittedly reasonable) conclusion about tumor control regardless. Consider changing the stated conclusions, waiting for longer duration of follow-up, or excluding cases with a shorter duration of follow-up.

With regards to postoperative treatment with gamma knife radiosurgery, the authors have clarified that no patients were treated prophylactically. What was the mean growth observed (and ranges of growth) that prompted GKRS in the 10 patients that received it? Specifically, what growth was seen for the patient that received GKRS at 4 months?

7. PLOS authors have the option to publish the peer review history of their article (what does this mean?). If published, this will include your full peer review and any attached files.

Reviewer #1: No

Reviewer #2: No

---

## [Author Response · Author response to Decision Letter 1]

7 Apr 2021

EDITOR SUGGESTIONS:

1. Regarding involvement of the mastoid segment: does this mean the mastoid segment of the facial nerve had vestibular schwannoma within it? How is this distinguished from facial nerve schwannoma? The pathology report alone cannot distinguish the nerve of origin.

• RESPONSE: Thank you very much for your question. All of our patients enrolled to our study had no preoperative facial palsy and diagnosed as huge tumor size >3 cm. That would be less possible for patients with huge size of facial nerve schwannoma who have no preoperative facial palsy in our experience. 

REVIEWER 1 COMMENTS:

Thank you for suggestion of further English Language proof-reading service. The manuscript has been carefully reviewed by an experienced editor whose first language is English again. We have corrected the grammatical errors in our revised manuscript. 

2. Lines 64-65: This sentence needs to be rephrased. There certainly are papers discussing IAC tumor removal. The authors need to make this sentence clearer. 

RESPONSE: Thank you for this suggestion. We have rephrased this sentence in our revised manuscript. 

3. Line 72: ‘Incidence’ should be ‘prevalence’.

RESPONSE: We have corrected the word in our revised manuscript. Thank you for help.

4. Lines 72-75: This sentence should be revised. Facial nerve palsy complicating surgery is awkwardly phrased.

RESPONSE: Thank you for this suggestion. We have rephrased this sentence in our revised manuscript.

5. Lines 86: Please consider removing this sentence. It is redundant, you have already stated the translab approach was used in the previous sentence.

RESPONSE: Thank you for this suggestion. We have removed this sentence because that was redundant after careful consideration.

6. Line 89-90: This sentence needs revision, either remove ‘and reports’ or change the sentence to avoid having two ‘and’s’.

RESPONSE: We have removed the words ‘and reports’ in our revised manuscript. Thank you for help.

7. Lines 107-109: Please add a figure showing how the measurements were made.

RESPONSE: We illustrated the measurement of tumor size using the maximal diameter of the extracanalicular component from the axial/coronal view of MRI scan as following. 

8. Line 114: ‘surgical findings of surgeons’ is awkward and redundant, should just be ‘intraoperative findings’ 

RESPONSE: Thank you for this suggestion. We have rephrased this sentence in our revised manuscript.

9. Line 125: ‘and other pathological types’ is vague and needs rewording

RESPONSE: We have rephrased this sentence as “All patients aged <20 years, neurofibromatosis type II (NF2), non-vestibular schwannoma, recurrent tumor, preoperative facial palsy grade III–VI, and mortalities were excluded from the study.” in our revised manuscript. Thank you for help.

10. Lines 127-130: Sentence needs to be rephrased. It is repetitive and awkwardly rephrased.

RESPONSE: Thank you for this suggestion. We have rephrased this sentence as “GKRS was arranged if regrowth of tumor remnant was identified from subsequent postoperative MRI by the senior neurosurgeon” in our revised manuscript.

11. I’m not sure why there are no line numbers in the paragraphs between lines 155 and 156, but the sentence with the word ‘half-invasion’ needs to be redone.

RESPONSE: We are sorry for this mistake and have added the line numbers in that paragraphs. Also, we have corrected this word to ‘half-canal involvement’ in our revised manuscript. Thank you for help. 

12. Line 173: “After then” should be changed

RESPONSE: Thank you for this suggestion. We have rephrased this sentence as “We checked MRI every 6 to 12 months afterwards to assess tumor control” in our revised manuscript.

13. Lines 198-204: This paragraph was already stated in the methods section, unclear why it is in the discussion again

RESPONSE: In this paragraph, we tried to explain that there was no iatrogenic nerve injury and we performed surgery delicately with intraoperative nerve monitoring. We agree with your suggestion that this paragraph was unclear and some redundant sentence. We have revised the text after careful consideration. 

14. Lines 247-250: This sentence is unclear and needs to be revised.

RESPONSE: We have rephrased this sentence as “Similar to previous reports, our results support the idea that near-total resection has greater therapeutic outcomes and a lower risk of recurrence. Nonetheless, postoperative follow-ups using serial MRI for the early detection of recurrence are necessary”. Thank you for this suggestion.

15. Lines 249: Please remove the word ‘alert’

RESPONSE: Thank you for this suggestion. We have rephrased this sentence mentioned as comment #13.

REVIEWER 2 COMMENTS:

1. The authors provided additional details regarding the statistical methods used. Given the multiple factors that likely affect surgical outcomes - both facial function and tumor control - I would have liked to see multivariate adjustment in the regressions to determine the significance of individual tumor or treatment factors of interest. If this was already done, it is not clear to the reader and should be clarified.

• RESPONSE: Thank you for this suggestion. Our results of predictive factors for facial palsy had been adjusted by multivariable linear regression initially. We are sorry for vague legends and have clarified that in our manuscript. We supplemented the statistical methods used as following. We performed statistical analyses for tumor control by the chi-square and Fisher’s exact tests due to categorical data. 

2. Additionally, I remain concerned that there is adequate duration of follow-up for all patients to draw the stated conclusion regarding tumor control. Consider that in some reports the average time to remnant tumor growth is over 1 year (Kasbekar AV, Adan GH, Beacall A, Youssef AM, Gilkes CE, Lesser TH. Growth Patterns of Residual Tumor in Preoperatively Growing Vestibular Schwannomas. J Neurol Surg B Skull Base. 2018;79(4):319-324. doi:10.1055/s-0037-1607421), yet some patients in the current study have duration of follow-up as short as 6 months. I commend that the authors have acknowledged this as a limitation, yet they make a (admittedly reasonable) conclusion about tumor control regardless. Consider changing the stated conclusions, waiting for longer duration of follow-up, or excluding cases with a shorter duration of follow-up. With regards to postoperative treatment with gamma knife radiosurgery, the authors have clarified that no patients were treated prophylactically. What was the mean growth observed (and ranges of growth) that prompted GKRS in the 10 patients that received it? Specifically, what growth was seen for the patient that received GKRS at 4 months?

• RESPONSE: Thank you for providing these insights. We arranged MRI scan within three days after surgery to be baseline of postoperative result. The case was treated with GKRS at 4 months because definite enlargement of residual tumor size was noted by the second time of postoperative MRI. We provide the immediate postoperative MRI and post-3 month MRI for supporting the reason we arranged GKRS at postoperative 4th month for the case. 

According to our medical charts, the case has no further recurrence after GKRS until now. We agree that it was really a limitation of our study with short following time and we mention this in the limitation section of the revised manuscript.

*more detail in attachment file"response to reviewer"

---

## [Decision Letter · Decision Letter 2]

3 Jun 2021

Impact of extent of internal acoustic meatus tumor removal using translabyrinthine approach for acoustic neuroma surgery

PONE-D-20-38944R2

Dear Dr. WANG,

We’re pleased to inform you that your manuscript has been judged scientifically suitable for publication and will be formally accepted for publication once it meets all outstanding technical requirements.

Kind regards,

Jennifer Alyono

Academic Editor

PLOS ONE

Additional Editor Comments (optional):

Reviewers' comments:

Reviewer's Responses to Questions

**Comments to the Author**

1. If the authors have adequately addressed your comments raised in a previous round of review and you feel that this manuscript is now acceptable for publication, you may indicate that here to bypass the “Comments to the Author” section, enter your conflict of interest statement in the “Confidential to Editor” section, and submit your "Accept" recommendation.

Reviewer #1: All comments have been addressed

Reviewer #2: All comments have been addressed

2. Is the manuscript technically sound, and do the data support the conclusions?

Reviewer #1: Yes

Reviewer #2: Yes

3. Has the statistical analysis been performed appropriately and rigorously? 

Reviewer #1: Yes

Reviewer #2: Yes

4. Have the authors made all data underlying the findings in their manuscript fully available?

Reviewer #1: Yes

Reviewer #2: Yes

5. Is the manuscript presented in an intelligible fashion and written in standard English?

Reviewer #1: Yes

Reviewer #2: Yes

6. Review Comments to the Author

Reviewer #1: (No Response)

Reviewer #2: I thank the authors for their earnest efforts to address my prior concerns and comments. Specifically, I appreciate the clarification of multivariable regression used in determining the effect of IAC dissection on facial function.

I continue to have concerns regarding the stated conclusions regarding tumor control due to inadequate duration of follow-up for some patients in the analyzed cohort. As suggested in my prior comments, the authors could have considered changing the stated conclusions, waiting for longer duration of follow-up, or performing a subgroup analysis on patients with adequate follow-up time. Short of that, the authors have addressed this as a limitation and readers should interpret these results critically and with caution.

7. PLOS authors have the option to publish the peer review history of their article (what does this mean?). If published, this will include your full peer review and any attached files.

Reviewer #1: No

Reviewer #2: **Yes: **Pedrom C. Sioshansi, MD

---

## [Editor Report · Acceptance letter]

16 Jul 2021

PONE-D-20-38944R2 

Impact of extent of internal acoustic meatus tumor removal using translabyrinthine approach for acoustic neuroma surgery 

Dear Dr. WANG:

I'm pleased to inform you that your manuscript has been deemed suitable for publication in PLOS ONE. Congratulations! Your manuscript is now with our production department. 

Kind regards, 

on behalf of

Dr. Jennifer Alyono 

Academic Editor

PLOS ONE